# Influence of Symbiotic Fermentation Broth on Regulating Metabolism with Gut Microbiota and Metabolite Profiles Is Estimated Using a Third-Generation Sequencing Platform

**DOI:** 10.3390/metabo13090999

**Published:** 2023-09-08

**Authors:** Chih-Yin Wu, Chun-Kai Huang, Wei-Sheng Hong, Yin-Hsiu Liu, Ming-Chi Shih, Jung-Chun Lin

**Affiliations:** 1Department of Family Medicine, Wan Fang Hospital, Taipei Medical University, Taipei City 116, Taiwan; 109004@w.tmu.edu.tw; 2Department of Laboratory Medicine, Wan Fang Hospital, Taipei Medical University, Taipei 116, Taiwan; 99421@w.tmu.edu.tw; 3Sagittarius Life Science Corporations, Taipei 115, Taiwan; robert.hong@naturalbiokey.com (W.-S.H.); frozenstock.liu@naturalbiokey.com (Y.-H.L.); richard@naturalbiokey.com (M.-C.S.); 4School of Medical Laboratory Science and Biotechnology, College of Medical Science and Technology, Taipei Medical University, New Taipei City 235, Taiwan; 5PhD Program in Medical Biotechnology, College of Medical Science and Technology, Taipei Medical University, New Taipei City 235, Taiwan; 6Pulmonary Research Center, Wan Fang Hospital, Taipei Medical University, Taipei 116, Taiwan

**Keywords:** ONT sequencing, gut microbiota, gut metabolite, symbiotic fermentation broth, LC-MS/MS, MetaCyc database

## Abstract

Overnutrition with a high-fat or high-sugar diet is widely considered to be the risk factor for various metabolic, chronic, or malignant diseases that are accompanied by alterations in gut microbiota, metabolites, and downstream pathways. In this study, we investigated supplementation with soybean fermentation broth containing saponin (SFBS, also called SAPOZYME) in male C57BL/6 mice fed a high-fat-fructose diet or normal chaw. In addition to the lessening of weight gain, the influence of SFBS on reducing hyperlipidemia and hyperglycemia associated with a high-fat-fructose diet was estimated using the results of related biological tests. The results of gut microbial profiling indicated that the high-fat-fructose diet mediated increases in opportunistic pathogens. In contrast, SFBS supplementation reprogrammed the high-fat-fructose diet-related microbial community with a relatively high abundance of potential probiotics, including *Akkermansia* and *Lactobacillus* genera. The metagenomic functions of differential microbial composition in a mouse model and enrolled participants were assessed using the PICRUSt2 algorithm coupled with the MetaCyc and the KEGG Orthology databases. SFBS supplementation exerted a similar influence on an increase in the level of 4-aminobutanoate (also called GABA) through the L-glutamate degradation pathway in the mouse model and the enrolled healthy population. These results suggest the beneficial influence of SFBS supplementation on metabolic disorders associated with a high-fat-fructose diet, and SFBS may function as a nutritional supplement for people with diverse requirements.

## 1. Introduction

Dietary intake has been widely considered to be a crucial factor in manipulating the constituents of the gut microbial community that are relevant to the healthy status of the host [1]. Recently, probiotic or prebiotic supplementation in symbiotic fermentation has been increasingly applied to manipulate the gut microenvironment in animal models or clinical practice with high-risk conditions [2]. *Lactobacillus plantarum* supplementation has been documented to strengthen the mucosal integrity or immune response towards opportunistic pathogens within the human gut with the production of short-chain fatty acids [3]. Indeed, the diverse biological activities of *Lactobacillus plantarum* could alter the gut ecosystem of individuals. The majority of prebiotics are classified as carbohydrate-based ingredients that are generated through the utilization of probiotics, of which their presence is beneficial to gut health [4]. Inulin, which belongs to the classification known as fructans, is a common prebiotic that can be isolated in the roots of different plants. The benefits of inulin have been demonstrated to control blood sugar with the absorption of calcium and magnesium [5] and to facilitate the propagation of probiotics in the human gut [6]. Nevertheless, the metabolism and production of most carbohydrate-based ingredients with dietary intake are varied and affected by the microbial community in each individual.

In this study, the beneficial influence of commercial SFBS was evaluated using a mouse model and a human trial. In summary, consumption of SFBS diminished the impact of a high-fat-fructose (HFF) diet on increases in fasting blood sugar, total triglycerides, and total cholesterol in experimental mice without renal and liver toxicity. Furthermore, SFBS supplementation reprogrammed the HFF-manipulated gut microenvironment to a healthy status within mice or the human gut.

## 2. Materials and Methods

### 2.1. Production of SFBS

SFBS (SAPONZYME) is a concentrated symbiotic fermentation broth provided by Sagittarius Life Science Co., Ltd. In brief, soybean powder was applied as the fermentation substrate with *Bifidobacterium*, *Streptococcus*, *Lactobacillus* genera, and *Saccharomyces*. The fermented products were incubated at 37 °C for 100 h and sterilized to eliminate microorganisms. The compounds in the metabolic profile of SFBS were identified by using non-targeted liquid chromatography mass spectrometry (LC-MS).

### 2.2. Ethics Statement

The animal experiment procedures executed in this study were reviewed and approved by the Institutional Animal Care and Use Committee (IACUC) of Taipei Medical University (approval no. LAC-2021-0511). All the experiments were conducted on animals in accordance with the Guide for the Care and Use of Laboratory Animals (Institute of Laboratory Animal Resources, Commission on Life Sciences, National Research Council) to minimize the suffering of experimental animals. The protocol regarding the recruitment of healthy participants and fecal sample collection was reviewed and approved by the Institutional Review Board of Taipei Medical University (approval no. N202103179). Healthy participants were enrolled in the Health Examination Center at Taipei Municipal WanFang Hospital. A questionnaire was used to evaluate the lifestyle of all participants.

### 2.3. Animal Model

C57BL/6 male mice (five weeks old) were purchased from the National Laboratory Animal Center (Taipei, Taiwan) and housed in the animal facility at the Agricultural Technology Research Institute (Hsinchu, Taiwan) at room temperature (22 ± 1 °C), 60% relative humidity, and a 12 h light-dark cycle (light on 8:00 AM). Before the experiments, the mice were acclimatized for a week to the environment and the diet prior to the experiment. Water and a standard laboratory rodent diet 5001 (Scott’s Distribution, Hudson, NH) or a high-fat-fructose (HFF) diet (Research Diet D12451, 24% energy from fat and 41% energy from fructose; New Brunswick, NJ, USA) were provided ad libitum. The mice were randomly assigned to 1 of the following 4 treatments for 8 weeks (*n* = 5): (1) standard diet, (2) standard diet supplemented with SFBS 700 mg/kg/day, (3) HFF diet, (4) HFF diet supplemented with SFBS 700 mg/kg/day. Feces and peripheral blood were collected every 4 weeks. All the mice were euthanized by 95% CO asphyxiation, and the related organs or tissues were collected immediately.

### 2.4. Analysis of Biochemical Variables

The fasting blood samples were collected at the onset of the experiment and every 4 weeks throughout the process. The biochemical variables, including blood sugar, total cholesterol (TC), triglycerides (TG), aspartate aminotransferase (AST), blood urea nitrogen (BUN), and creatinine (CRE), were examined using a dry chemistry analyzer (Catalyst One, IDEXX Laboratories, Westbrook, ME, USA).

### 2.5. Isolation of Total Genomic DNA in the Fecal Samples

The fecal samples were preserved using Fecal Collection tubes (Zymo Research, Irvine, CA, USA) to diminish the growth of bacteria. Total genomic DNA was extracted by using a stool DNA extraction kit (Geneaid, Taipei, Taiwan) in accordance with the manufacturer’s instructions. The quantity of purified DNA samples was estimated by using a fluorometric assay (GeneCopoeia, Rockville, MD, USA).

### 2.6. Profiling of Gut Microenvironment

The gut microbial community was classified using an Oxford Nanopore sequencing platform (ONT, Oxford, UK). In brief, 10 ng total gDNA of each sample was applied for barcoding of 16S ribosomal RNA (or 16S rRNA) gene sequences using the SQK-16S024 kit (ONT) in accordance with the manufacturer’s protocol. Two ng of barcoded DNA for each sample was pooled, ligated with the adapter, and sequenced on a MinION MK1C sequencer and flow cells (FLO-MIN106D R9.4.1, MinION instrument, ONT). The mean read number was 100,000 per sample to meet a sufficient reading depth. The raw reads were quality checked and clustered to operational taxonomic units (OTUs) with 22,596 16S rRNA reference sequences curated from the Bacterial 16S Ribosomal RNA RefSeq Targeted Loci Project (Accession No. PRJNA33175, NCBI) with 97% similarity by using the CLC genomics workbench (v23.0.4) composed of the Minimap2 program and the Microbial Genomics Module (v23.0.1, Qiagen, CLC bio, Aarhus, Denmark). An OTU table was applied to construct the phylogenetic tree, using the MUSCLE 2.0 and Maximum Likelihood Phylogeny tools (CLC Genomics Workbench), which was subjected to estimate the alpha diversity metrics and the inter-sample dissimilarity (beta diversity). The differential abundance of identified OTUs among all groups was assessed using the linear discriminant analysis (LDA) effect size (LEfSe) method via the website algorithm (https://huttenhower.sph.harvard.edu/galaxy/root, accessed on 1 August 2023) with the default setting. The inter-sample difference of identified taxa was statistically convincing with a *p*-value < 0.05 and an LDA score (log10) >3 or <−3. Potential relevance between the gut microbial community and the metabolic pathway was assessed by using the Phylogenetic Investigation of Communities by Reconstruction of Unobserved States (PICRUSt2) platform adopted to the KEGG Ortholog database (KO, https://www.genome.jp/kegg/ko.html, accessed on 1 August 2023), the Enzyme Commission (EC), and the MetaCycdatabase (https://metacyc.org/, accessed on 1 August 2023).

### 2.7. Metabolites of SFBS Are Profiled Using an Untargeted LC-MS/MS Analysis

The extraction of fermented metabolites within SFBS was commissioned to the BIOTOOLS Co., Ltd. (Taipei, Taiwan). In brief, 200 μL SFBS was mixed with 600 μL 100% methanol. The mixture was homogenized with sonication and incubated at −20 °C for 1 h. The mixture was centrifugated at 12,000 rpm for 15 min, and the supernatant was transferred to a sample vial. Then, 10 μL extract of each sample was injected into a vanquish-focused ultra-performance liquid chromatography (UPLC) system coupled with an Orbitrap Elite Mass Spectrometry system (Thermo Fisher Scientific, San Jose, CA, USA). The parameters were set as follows: The 2.1 × 100 mm Acquity BEH 1.7 μm C18 column (Waters) was applied at 40 °C. The binary mobile phase was composed of deionized water containing 0.1% formic acid as solvent A and LC-MS grade acetonitrile with 0.1% formic acid as solvent B. The flow rate was set at 0.25 mL/min with a linear gradient elution for 15 min. The percentage of solvent B was linearly increased from 0% to 100%, kept constant for 3 min, and decreased to 0% in the last 1 min. A blank injection was carried out after the sample injection to prevent carryover effects, and a QC injection was executed every five samples to normalize the peak area. MS full scan was conducted in profile mode at 60,000 resolutions, followed by data-dependent MS2 scans at 15,000 resolutions, and the scan range was set from 70 to 1000 *m*/*z*. The scanned data were converted to the mzXML format using the ProteoWizard software, which was subjected to annotation. The converted results were assessed using an in-house XCMS program for peak detection, extraction, alignment, and integration (BIOTOOLS). An in-house MS2 database (BiotreeDB, BIOTOOLS) was used for the annotation of scanned fragments.

### 2.8. Statistical Analysis

The statistical significance of the analytic results was shown in the mean ± standard error (SEM). Continuous variables of the analytic results were estimated by using a one-way analysis of variance (ANOVA) combined with Tukey’s multiple comparison post hoc test. The difference was identified as statistically significant with a *p*-value less than 0.05. * *p* < 0.05, ** *p* < 0.01, and *** *p* < 0.005.

## 3. Results

### 3.1. The Metabolite Profile of SFBS Is Assessed Using the LC-MS/MS Approach

Prior to the in vivo or in vitro functional analysis, the compound profile of SFBS was classified by using the UPLC-MS/MS approach. Under the positive-charge mode, the relatively high abundances of 12 metabolites, such as soyasaponin and dopamine, were identified in SFBS as compared to those of fundamental broth (Table 1). The relatively high abundances of 13 metabolites, including ornithine or epigallocatechin, were classified in SFBS under the negative-charge mode (Table 2). The metabolite profile of SFBS suggests its potential benefit to health.

### 3.2. SFBS Supplementation Diminishes the Impact of a High-Fat-Fructose Diet on Biochemical Variables in Serum

The influence of SFBS supplementation on healthy status was first evaluated using the mouse model with vital signs and biochemical variables. As shown in Figure 1, the steady gain in body weight of the four groups was monitored throughout the experiment, and SFBS supplementation (Figure 1a, yellow line) lessened the influence of the HFF diet (Figure 1a, gray line) on increasing body weight of experimental mice. Relatively high levels of biochemical variables, including blood sugar, total triglyceride, and total cholesterol, were noted in the HFF groups (Figure 1b–d, gray bar) as compared to those of mice fed the HFF diet and SFBS, 8 weeks post the onset of the experiment (Figure 1b–d, yellow bar). These results suggest the effect of SFBS supplementation on maintaining the metabolic homeostasis of experimental mice fed the HFF diet.

### 3.3. SFBS Supplementation Reprograms HFF Diet-Manipulated Gut Microenvironment

A high-fat or high-glucose diet has been widely considered to be a high-risk factor involved in the occurrence of diverse aging, chronic, and malignant diseases [1,7], which might alter the homeostasis of a gut microbial community. In this study, the classified I table was subjected to the diversity assessment. As shown in Figure 2A, the relatively low Simpson’ s dominance index in the HFF diet group (Figure 2A, dark blue box) indicated poor evenness in the gut microbial community as compared to that of the normal diet group, whereas SFBS supplementation restored the HFF-manipulated evenness in gut microbiota (Figure 2A, pink box). The results of the PERMANOVA assay indicated the dissimilarity between the gut microbiota in the normal diet group and the HFF diet group (Figure 2B, green and orange dots, *p* = 0.0012). SFBS supplementation diminished the impact of the HFF diet, leading to alterations in the gut microbial community as compared to that of the normal diet group (Figure 2B, green and light blue dot, *p* = 0.0088).

In summary, intake of the HFF diet resulted in increases in specific OTUs, such as *Akkermansia*, *Romboutsia*, and *Parabacteroides* genera, with a concomitant decrease in *Lactobacillus* genera in the gut microbial community as compared to those of the normal diet group (Table 3, upper panel). By contrast, the alterations in these OTUs were less robust in the HFF diet group supplemented with SFBS as compared to the normal diet group (Table 3, lower panel).

The influence of HFF diet-manipulated gut microbiota on the metabolic profile was imputed by deploying the PICRUSt2 platform coupled with the MetaCyc database, the Enzyme Commission (EC) pathway, or the KEGG Orthology database. The analytic results indicated that the intake of the HFF diet potentially led to increases in the generation of 4-aminobutanoate (Table 4, 10.07 folds, also known as gamma-aminobutyric acid) and histamine (Table 4, 6.84 folds) as compared to the normal diet group. Nevertheless, the synchronous SFBS supplementation with the HFF diet resulted in a further high level of 4-aminobutanoate (Table 4, 53.42 folds) with a concomitant decrease in histamine production (Table 4, 4.97 folds). Taking these results together, SFBS supplementation lessened the impact of the HFF diet on disturbing gut homeostasis.

### 3.4. SFBS Supplementation Reprograms the Gut Microenvironment of Enrolled Participants

In this study, the influence of SFBS was further validated by conducting a human trial with the recruitment of 10 healthy participants. As shown in Figure 3, SFBS supplementation resulted in an increased abundance of normal flora in the gut (Figure 3A), which was relevant to the less evenness of the OTU profile in the post-SFBS group with the downregulated Simpson’s dominance index (Figure 3B, blue bar). Although SFBS supplementation exhibited no influence on the dissimilarity between these two groups (Figure 3C, *p* = 0.0003), the relative abundances of colorectal cancer (CRC) or irritable bowel syndrome (IBS)-related OTUs, including *Eubacterium rectale*, *Roseburia intestinalis*, and *Clostridium saudiense*, were diminished in the participants supplemented with SFBS (Table 5). The OTU profile was further subjected to predict the metabolic pathway through the MetaCyc algorithm. The analytic results indicated upregulated activity associated with the production of 4-aminobutanoate, 1-amino-2-propanol O-2-phosphate, and catechol, in the gut microenvironment of recruited participants with SFBS supplementation (Table 6), which were results that were consistent with those of the mouse model fed with SFBS (Table 4). Taken together, SFBS supplementation exerted a consistent influence on manipulating the gut microbial communities and metabolite profiles in distinct species.

## 4. Discussion

High-fat-sugar dietary intake has been widely classified as a high-risk diet associated with the occurrence of diverse diseases with elevated levels of serum triglyceride, total cholesterol, and blood sugar [1,7]. Saponin compound, identified in SFBS using an untargeted LC-MS/MS approach in this study, has been demonstrated to exhibit a pharmacological impact on modulating immune response and metabolic homeostasis [8]. The identified compounds, such as eugenol, homocarnosine, genistein, dopamine, ornithine, and distinct short-chain fatty acids, within SFBS have been demonstrated to exert pharmacological effects on diverse physiological events [9,10,11,12,13,14]. These results suggest that the SFBS could function as a nutritional supplement for populations with diverse requirements.

In addition to the biochemical parameters, the relevance between a long-term HFF diet and the gut microbial community has been recently and widely pursued. The differential influence between a high-fat or high-glucose diet on reprogramming gut microbiota has been noted. In summary, increases in the relative abundance of *Firmicutes* with concomitant decreases in the *Bacteroides* and *Akkermansia* have mostly been classified in mouse models with high-fat diets (from 45% to 75%) [15,16]. In contrast, supplementation of a high-glucose or high-fructose diet has resulted in an elevated abundance of *Akkermansia muciniphila* which has been proposed to exert a protective effect against obesity, diabetes, or other metabolic disorders [17,18,19]. In this study, increases in the relative abundance of *Akkermansia*, *Romboutsia*, *Parabacteroides*, and *Clostridium* genera with a concomitant decrease in the relative level of *Lactobacillus* genera were classified in the gut microbial community of the HFF group, which was consistent with other reports [20]. In a previous study, supplementation with the non-edible components of green asparagus, which include xylose, inulin, flavonoids, and saponins, was demonstrated to facilitate the growth of *Lactobacillus* and *Bifidobacteria* genera with the execution of an in vitro culture model [21]. Additionally, Synbiotic supplement composed of probiotics, including *Lactobacillus* and *Bifidobacteria* genera, and prebiotics has been demonstrated to exhibit a statistically significant effect on decreases in obesity-related biomarkers in conducted clinical trials [22]. These studies have consistently illustrated the relevance between gut microenvironment and host health. In this study, the presence of saponin within SFBS supplementation diminished the impact of an HFF diet on reducing the relative levels of probiotic bacteria in a mouse model. Moreover, SFBS supplementation composed of variable compounds may lessen the influence of an HFF diet on disturbing the composition of gut microbiota via a more complex but solid network, which is beneficial to maintain the hemostasis of the gut microenvironment of a host. Nevertheless, the influence of SFBS supplementation on the gut microbial community associated with distinct dietary types by using animal models or human trials is worthy of further investigation.

Although there have been considerable efforts to identify or to pursue disease-relevant OTUs or the enterotype to function as the biomarker [23], the gut microbiota-driven metabolic network and generated metabolites have been shown to have a direct effect on manipulating the gut environment of host physiology [24]. To fill the knowledge gap or diminish the ambiguous results of OTU classification, the PICRUSt2 platform coupled with the Enzyme Commission, and the MetaCyc and KEGG Orthology databases have constituted a widely used workflow for predicting the metagenomic or metabolic function of the microbial community [25]. The analytic results have indicated relatively active metabolism pathways with the gut microbiota identified in high-fat or high-sugar mouse models, which have suggested the credibility of PICRUSt2-based predictions [26,27]. Except for energy metabolism, an HFF diet has been estimated to enhance diverse enzymatic activities of the gut microbial community, such as the synthesis of 4-aminobutanoate, histamine, and bacterial toxin in this study, whereas SFBS supplementation has exhibited an opposite effect on manipulating HFF-activated 4-aminobutanoate or histamine synthesis. Moreover, SFBS supplementation has exerted a consistent influence on similar pathway profiles associated with gut microbiota in mouse models and human trials. The targeted LC-MS/MS approach is worthy of conduction to further validate these analytic results.

## 5. Conclusions

To conclude, synchronous SFBS supplementation generated with multiple probiotics and soybean lessened the increases in weight gain and biochemical parameters of energy metabolism in the HFF diet group. Intervention with SFBS intake diminished the impact of the HFF diet on manipulating the gut microbial community and corresponding enzymatic activity and pathway with animal experiments. Furthermore, SFBS supplementation exhibited a consistent effect on reprogramming the gut microenvironment in experimental mice and enrolled participants without adverse events. These results suggest the benefit and safety of SFBS as a nutritional supplement.

## Figures and Tables

**Figure 1 metabolites-13-00999-f001:**
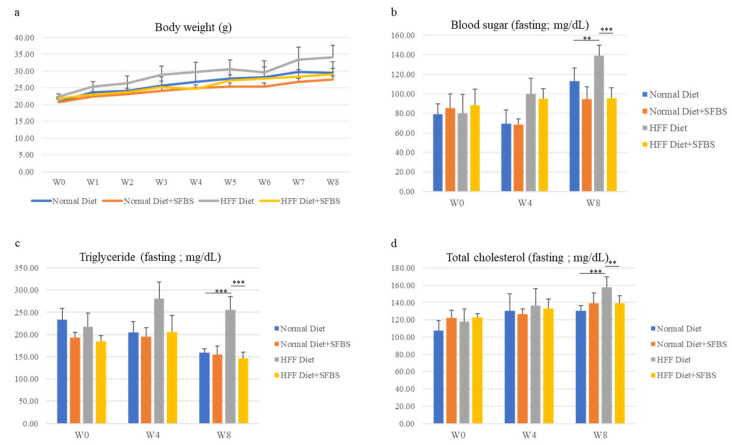
The body weight (**a**), sera levels of fasting glucose (**b**), fasting triglyceride (**c**), and total cholesterol (**d**) of each experimental group were monitored at the onset of the experiment (W0), fourth (W4), and eighth week (W8). *** *p* < 0.001 and ** *p* < 0.005.

**Figure 2 metabolites-13-00999-f002:**
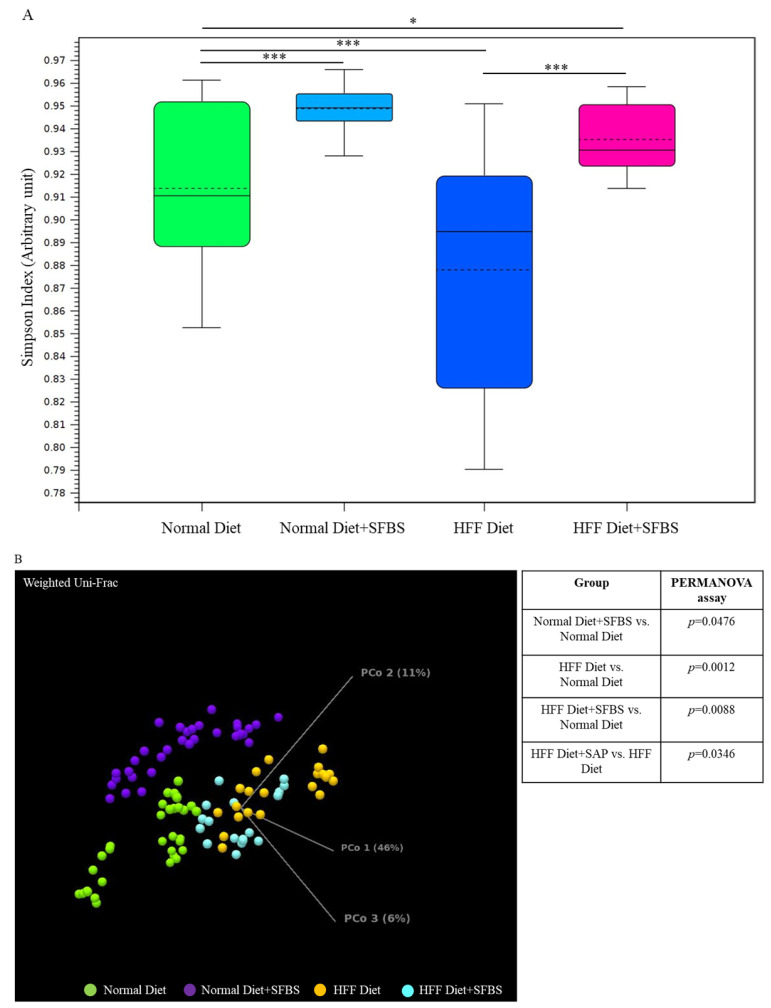
Dietary intake or SFBS supplementation leads to alterations in the gut microbiota of experimental mice: (**A**) HFF diet or SFBS supplementation exhibited the opposite effect on changing the evenness (Simpson’s index) of the gut microbial community; (**B**) HFF diet or SFBS supplementation enhanced the dissimilarity in the IOTU profile compared to the normal diet group, which was estimated by using the Weighted Unifrac principal component analysis (PCoA) and PERMANOVA index. *** *p* < 0.001 and * *p* < 0.01.

**Figure 3 metabolites-13-00999-f003:**
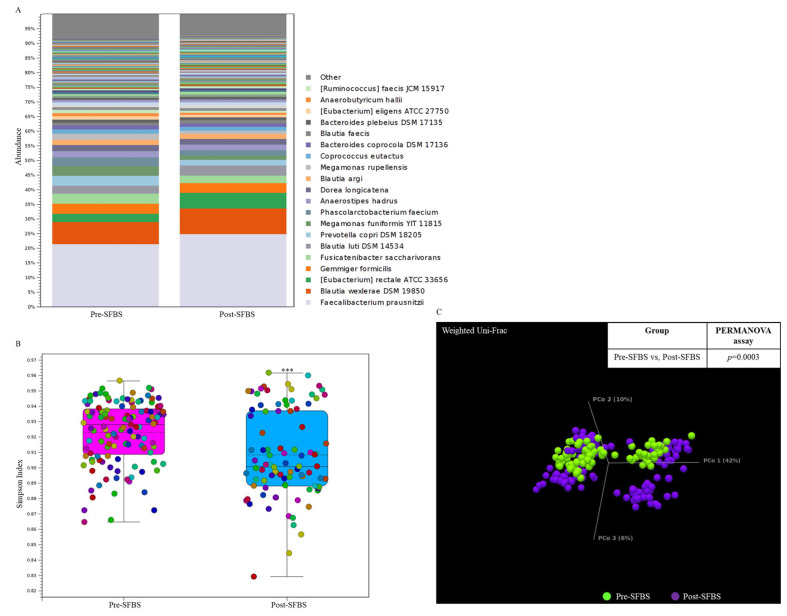
SFBS supplementation reprograms the gut microbial community in recruited participants. (**A**) The operational taxonomy unit (OTU) profile with long-read sequencing in the enrolled participants was analyzed by using the ONT sequencing platform and presented in a stacked bar chart. (**B**) Supplementation of SFBS mediated the reduced evenness of gut microbial community with low Simpson’ s index. (**C**) SFBS supplementation mediated the dissimilarity in OTU profile of the same participants, which is evaluated by using Weighted Unifrac principal component analysis (PCoA) coupled with PERMANOVA index. (***, *p* < 0.001).

**Table 1 metabolites-13-00999-t001:** Statistics of identified metabolites in positive-charge mode.

ID	CpdName	Score	KEGGID	HMDBID	Microbe	Rtmed	Mzmed	Exp Mean	Ctrl Mean	VIP	*p*-Value	Fold Change
FT0338	Eugenol	0.99059	C10453	HMDB0005809	NA	246.5537	128.0419	1.36 × 10^−2^	2.18 × 10^−5^	1.253382	0.017216	624.381
FT6284	Fasciculic acid B	0.90713	NA	HMDB0036438	NA	426.4757	795.4509	1.5 × 10^−5^	6.05 × 10^−8^	1.425796	0.006844	247.4448
FT0799	Homocarnosine	0.78974	C00884	HMDB0000745	YES	318.4291	174.1119	1.13 × 10^−3^	1.29 × 10^−5^	1.240513	0.013296	87.37652
FT4987	Moclobemide	0.80195	NA	HMDB0015302	NA	266.6079	479.2484	2.9 × 10^−4^	9.02 × 10^−5^	0.399071	0.011479	32.17849
FT1164	Glycylleucine	0.58063	C02155	HMDB0000759	NA	84.13725	203.0521	2.41 × 10^−3^	9.7 × 10^−5^	1.198751	0.005106	24.86357
FT6289	Soyasaponin I	0.87364	C08983	HMDB0034649	NA	407.55	797.4662	7.55 × 10^−4^	4.02 × 10^−5^	1.02781	0.000364	18.79097
FT2334	Genistein	0.99973	C06563	HMDB0003217	YES	341.9878	271.0593	4.5 × 10^−4^	3.1 × 10^−5^	1.352531	0.000707	14.54516
FT1837	L-Glutamine	0.91368	C00064	HMDB0000641	YES	241.1555	244.1284	2.97 × 10^−4^	2.35 × 10^−5^	1.37705	0.023164	12.63435
FT4728	Phlorizin	0.69916	C01604	HMDB0036634	YES	407.3074	448.243	5.23 × 10^−5^	4.62 × 10^−6^	1.351436	0.007354	11.31761
FT1557	Altretamine	0.73286	NA	HMDB0014631	NA	417.2145	228.1952	2.19 × 10^−3^	2.16 × 10^−4^	1.198833	0.027437	10.12372
FT0340	Dopamine	0.96353	C03758	HMDB0000073	YES	296.1454	332.2169	2.17 × 10^−4^	4.6 × 10^−5^	1.336598	0.037817	4.722736
FT0495	1,2,3,4-Tetrahydro-beta-carboline	0.99943	NA	HMDB0012488	NA	287.099	144.0803	6.98 × 10^−3^	2.68 × 10^−3^	1.166997	0.046653	2.606954

**Table 2 metabolites-13-00999-t002:** Statistics of identified metabolites in negative-charge mode.

ID	CpdName	Score	KEGGID	HMDBID	Microbe	Rtmed	Mzmed	Exp Mean	Ctrl Mean	VIP	*p*-Value	Fold Change
FT1381	Isocitric acid	0.96879	C00311	HMDB0000193	YES	94.38789	353.072	0.020077	6.85 × 10^−7^	1.280033	0.045768	29325.62
FT0754	DL-O-Phosphoserine	0.64298	C01005	HMDB0001721	YES	54.73537	259.0221	0.003051	7.57 × 10^−5^	1.06287	0.008156	40.28009
FT1176	Aspalathin	0.96254	NA	HMDB0030851	NA	331.2802	331.118	0.001473	3.99 × 10^−5^	1.246585	0.021972	36.96136
FT0090	Malic acid	0.94492	C00149	HMDB0000156	YES	68.6133	133.0143	0.026847	8.41 × 10^−4^	1.154805	0.021612	31.92659
FT0234	Rosmarinic acid	0.86928	C01850	HMDB0003572	YES	63.7528	179.0562	0.044586	1.57 × 10^−3^	1.276599	0.025802	28.32409
FT2818	Phenytoin	0.89564	C07443	HMDB0014397	NA	94.07186	503.1612	0.000834	4.9 × 10^−5^	1.278964	0.002793	17.00339
FT1535	Ornithine	0.7267	C00077	HMDB0000214	YES	236.4346	515.1625	0.000537	4.45 × 10^−5^	1.14777	0.032421	12.06561
FT0207	L-Leucine	0.99821	C00123	HMDB0000687	YES	317.1682	172.0978	0.001018	8.7 × 10^−5^	1.137649	0.019915	11.69737
FT1500	(-)-Epigallocatechin	0.99869	C12136	HMDB0038361	YES	376.3322	253.0502	0.000785	8.39 × 10^−5^	1.181254	0.010312	9.351916
FT0017	Succinic acid	0.93795	C00042	HMDB0000254	YES	70.84436	101.0245	0.000561	1.22 × 10^−4^	1.122522	0.042969	4.60625
FT0132	Cortisone	0.61173	C00762	HMDB0002802	YES	331.8418	147.045	0.011588	2.63 × 10^−3^	1.233049	0.000266	4.406416
FT2017	Citric acid	0.99132	C00158	HMDB0000094	YES	72.08018	405.0281	0.000482	1.31 × 10^−4^	1.035968	0.029749	3.676637
FT0078	Genipin	0.83861	C09780	HMDB0035830	NA	314.9441	129.0557	0.001566	5.69 × 10^−4^	1.150752	0.000124	2.751451

**Table 3 metabolites-13-00999-t003:** Statistics of identified OTUs with differential abundance in the mouse model.

**HFF Diet vs. Normal Diet**				
**Name**	**Fold Change**	***p*-Value**	**FDR *p*-Value**	**Bonferroni**
*Akkermansia muciniphila* ATCC BAA-835; NR_042817.1	151.15	3.61 × 10^−23^	2.36 × 10^−20^	4.73 × 10^−20^
*Romboutsia timonensis*; NR_144740.1	104.99	3.03 × 10^−18^	1.32 × 10^−15^	3.96 × 10^−15^
*Akkermansia muciniphila*; NR_074436.1	96.59	1.09 × 10^−15^	3.55 × 10^−13^	1.42 × 10^−12^
*Akkermansia glycaniphila*; NR_152695.1	74.95	2.29 × 10^−12^	4.99 × 10^−10^	2.99 × 10^−9^
*Romboutsia ilealis*; NR_125597.1	59.95	1.96 × 10^−11^	3.67 × 10^−9^	2.57 × 10^−8^
*Shigella boydii*; NR_104901.1	25.39	1.02 × 10^−6^	9.57 × 10^−5^	1.34 × 10^−3^
*Luteolibacter gellanilyticus*; NR_158117.1	20.04	1.54 × 10^−6^	1.26 × 10^−4^	2.01 × 10^−3^
*Paraclostridium benzoelyticum*; NR_148815.1	19.27	2.20 × 10^−6^	1.69 × 10^−4^	2.87 × 10^−4^
*Parabacteroides goldsteinii* DSM 19448 = WAL 12034; NR_113076.1	10.45	2.03 × 10^−28^	2.66 × 10^−25^	2.66 × 10^−25^
*Eubacterium coprostanoligenes*; NR_104907.1	8.62	2.27 × 10^−5^	1.45 × 10^−3^	0.03
*Clostridium scindens*; NR_028785.1	4.31	1.66 × 10^−7^	1.67 × 10^−5^	2.17 × 10^−4^
*Parabacteroides chongii*; NR_165699.1	4.29	5.04 × 10^−6^	3.47 × 10^−4^	6.60 × 10^−3^
*Lactobacillus faecis*; NR_114391.1	−2.78	4.25 × 10^−8^	6.18 × 10^−6^	5.56 × 10^−5^
*Lactobacillus murinus*; NR_112689.1	−3.44	5.44 × 10^−10^	8.89 × 10^−8^	7.11 × 10^−7^
*Lactobacilpodemedemi*; NR_112752.1	−4.53	3.33 × 10^−6^	2.42 × 10^−4^	4.35 × 10^−4^
**HFF Diet + SFBS vs. Normal Diet**				
**Name**	**Fold Change**	***p*-Value**	**FDR *p*-Value**	**Bonferroni**
*Romboutsia timonensis*; NR_144740.1	35.93	5.54 × 10^−19^	1.81 × 10^−16^	7.25 × 10^−16^
*Akkermansia muciniphila* ATCC BAA-835; NR_042817.1	31.89	2.03 × 10^−13^	2.41 × 10^−11^	2.66 × 10^−10^
*Romboutsia ilealis*; NR_125597.1	29.72	2.55 × 10^−10^	1.85 × 10^−8^	3.34 × 10^−7^
*Akkermansia muciniphila*; NR_074436.1	30.81	1.23 × 10^−8^	6.70 × 10^−7^	1.61 × 10^−5^
*Paraclostridium benzoelyticum*; NR_148815.1	24.82	2.88 × 10^−7^	1.18 × 10^−5^	3.77 × 10^−5^
*Akkermansia glycaniphila*; NR_152695.1	19.69	4.64 × 10^−6^	1.52 × 10^−4^	6.08 × 10^−3^
*Eubacterium coprostanoligenes*; NR_104907.1	15.2	4.06 × 10^−8^	1.97 × 10^−6^	5.31 × 10^−5^
*Parabacteroides goldsteinii* DSM 19448 = WAL 12034; NR_113076.1	9.48	9.00 × 10^−24^	9.36 × 10^−21^	1.18 × 10^−20^
*Lactobacillus faecis*; NR_114391.1	−2.66	1.71 × 10^−6^	6.21 × 10^−5^	2.23 × 10^−3^
*Lactobacillus murinus*; NR_112689.1	−3.11	1.38 × 10^−7^	6.02 × 10^−6^	1.81 × 10^−4^
*Mucispirillum schaedleri*; NR_042896.1	−3.89	1.94 × 10^−5^	5.78 × 10^−4^	0.03

**Table 4 metabolites-13-00999-t004:** Statistics of the HFF diet-modulated metabolic pathway.

**HFF Diet vs. Normal Diet**						
**Pathway**	**MetaCyc ID**	**Min. Solution**	**Confidence**	**Coverage**	**Fold Change**	**Metabolite**
tetrahydro pteridine recycling	PWY-8099	TRUE	0.5	1	17.68	tetrahydrobiopterin
glutathionylspermidine biosynthesis	PWY-4121	TRUE	1	1	13.15	glutathionylspermidine
phenylmercury acetate degradation	P641-PWY	TRUE	1	1	13	catechol
uracil degradation III	PWY0-1471	TRUE	0.53	0.67	12	3-hydroxypropanoate
sulfoquinovose degradation I	PWY-7446	TRUE	1	0.8	11.78	dihydroxypropane-1-sulfonate
L-arginine degradation II (AST pathway)	AST-PWY	TRUE	1	1	11.59	glutamate
putrescine degradation IV	PWY-2	TRUE	0.56	0.67	10.07	4-aminobutanoate (GABA)
adenine and adenosine salvage V	PWY-6611	TRUE	0.52	0.67	8.4	ribosyl hypoxanthine monophosphate
enterobacterial common antigen biosynthesis	ECASYN-PWY	TRUE	1	0.67	7.7	enterobacterial common antigen
cinnamate and 3-hydroxycinnamate degradation to 2-hydroxypentadienoate	PWY-6690	TRUE	0.5	1	7.22	2-hydroxypenta-2,4-dienoate
histamine biosynthesis	PWY-6173	TRUE	1	1	6.84	histamine
aerobactin biosynthesis	AEROBACTINSYN-PWY	TRUE	1	1	6.33	aerobactin
aminopropanol phosphate biosynthesis I	PWY-5443	TRUE	1	1	6.05	1-amino-2-propanol O-2-phosphate
**HFF Diet + SFBS vs. Normal Diet**						
**Pathway**	**MetaCyc ID**	**Min. Solution**	**Confidence**	**Coverage**	**Fold Change**	**Metabolite**
glutathionylspermidine biosynthesis	PWY-4121	TRUE	1	1	81.6	glutathionylspermidine
sulfoquinovose degradation I	PWY-7446	TRUE	1	0.8	73.07	dihydroxy propane-1-sulfonate
L-arginine degradation II (AST pathway)	AST-PWY	TRUE	1	1	64.36	glutamate
uracil degradation III	PWY0-1471	TRUE	0.53	0.67	62.66	3-hydroxy propanoate
putrescine degradation IV	PWY-2	TRUE	0.54	0.67	53.42	4-aminobutanoate (GABA)
enterobacterial common antigen biosynthesis	ECASYN-PWY	TRUE	1	0.67	39.35	enterobacterial common antigen
cinnamate and 3-hydroxycinnamate degradation to 2-hydroxypentadienoate	PWY-6690	TRUE	0.52	1	38.31	2-hydroxypenta-2,4-dienoate
aerobactin biosynthesis	AEROBACTINSYN-PWY	TRUE	1	1	3.76	aerobactin
Adenine and adenosine salvage V	PWY-6611	TRUE	0.55	1	35.9	ribosyl hypoxanthine monophosphate
phenylmercury acetate degradation	P641-PWY	TRUE	1	1	33.05	catechol
tetrahydro pteridine recycling	PWY-8099	TRUE	0.52	1	13.51	tetrahydrobiopterin
aminopropanol phosphate biosynthesis I	PWY-5443	TRUE	1	1	8.7	1-amino-2-propanol O-2-phosphate
histamine biosynthesis	PWY-6173	TRUE	1	1	4.97	histamine

**Table 5 metabolites-13-00999-t005:** Statistics of identified OTUs with SFBS supplementation in the human trial.

Post-SFBS Group vs. Pre-SFBS Group				
Name	Fold Change	*p*-Value	FDR *p*-Value	Bonferroni
*Phascolarctobacterium succinatutens*; NR_112902.1	3.37	8.37 × 10^−5^	2.35 × 10^−4^	6.34 × 10^−4^
*Ruminococcus faecis* JCM 15917; NR_116747.1	2.11	3.27 × 10^−5^	3.54 × 10^−3^	0.02
*Phascolarctobacterium faecium*; NR_026111.1	1.81	2.23 × 10^−6^	3.38 × 10^−4^	1.69 × 10^−3^
*Fusicatenibacter saccharivorans*; NR_114326.1	1.68	1.69 × 10^−5^	2.14 × 10^−3^	0.01
*Eubacterium rectale* ATCC 33656; NR_074634.1	−1.79	1.33 × 10^−16^	1.01 × 10^−13^	1.01 × 10^−13^
*Roseburia intestinalis* L1-82; NR_027557.1	−2	4.31 × 10^−5^	4.08 × 10^−3^	0.03
*Clostridium saudiense*; NR_144696.1	−3.34	1.96 × 10^−6^	3.38 × 10^−4^	1.49 × 10^−3^

**Table 6 metabolites-13-00999-t006:** Statistics of SFBS-modulated metabolic pathway in the human trial.

Post-SFBS vs. Pre-SFBS						
Pathway	MetaCyc ID	Min. Solution	Confidence	Coverage	Fold Change	Metabolite
sulfoacetaldehyde degradation III	PWY-6718	TRUE	1	1	2.87	isethionate
L-arginine degradation XII	PWY-7523	TRUE	0.77	0.75	2.34	4-aminobutanoate
cellulose and hemicellulose degradation (cellulolosome)	PWY-6784	TRUE	1	0.73	2.34	polysacharide
vanillin and vanillate degradation II	PWY-7098	TRUE	1	0.5	2.34	protocatechuate
1,4-dihydroxy-6-naphthoate biosynthesis II	PWY-7371	TRUE	0.55	0.6	2.11	dihydroxyacetone
aminopropanol phosphate biosynthesis I	PWY-5443	TRUE	1	0.5	1.87	1-amino-2-propanol O-2-phosphate
gallate degradation I	GALLATE-DEGRADATION-II-PWY	TRUE	1	0.75	1.84	oxaloacetate
retinol biosynthesis	PWY-6857	TRUE	0.72	0.62	1.7	trans-retinol
benzoate degradation I (aerobic)	PWY-2503	TRUE	1	0.5	1.67	catechol
4-methyl catechol degradation (ortho cleavage)	PWY-6185	TRUE	0.58	0.54	1.67	acetyl-CoA
methiin metabolism	PWY-7614	TRUE	0.64	0.54	1.64	pyruvate
pyruvate fermentation to propanoate I	P108-PWY	TRUE	1	0.57	1.57	propanoate

## Data Availability

The data presented in this study are available on request from the corresponding author due to privacy restrictions.

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
