# Peer review of "Influence of Symbiotic Fermentation Broth on Regulating Metabolism with Gut Microbiota and Metabolite Profiles Is Estimated Using a Third-Generation Sequencing Platform"

_metabolites, 2023, doi:10.3390/metabo13090999_

Round 1

Reviewer 1 Report

Article "Influence of symbiotic fermentation broth on regulating me-tabolism with gut microbiota-metabolite profiles is estimated using third generation sequencing platform" (Authors: Chih-Yin Wu and at. all). 

In this paper, the authors investigate the impact of saponin supplementation on health status: changes in metabolic homeostasis depending on the intestinal ecosystem/diets. The article corresponds to the subject of the journal "Metabolites ".  The article is well structured, written in a clear and understandable language, the conclusions are logical, the literature corresponds to the stated topic.  The main conclusion: supplementation of SFBS lessened the influence of HFF diet on disturbing the composition of gut microbiota, which might be beneficial to maintain the hemostasis of the gut microenvironment.  As such, there are no comments on the work. However, all experiments are done on mice. Therefore, I would like to see the authors' reasoning regarding how the established patterns will correspond to a person.

I recommend publishing your work.

Author Response

Response to Reviewer 1's comments

In this paper, the authors investigate the impact of saponin supplementation on health status: changes in metabolic homeostasis depending on the intestinal ecosystem/diets. The article corresponds to the subject of the journal "Metabolites ".  The article is well structured, written in a clear and understandable language, the conclusions are logical, the literature corresponds to the stated topic.  The main conclusion: supplementation of SFBS lessened the influence of HFF diet on disturbing the composition of gut microbiota, which might be beneficial to maintain the hemostasis of the gut microenvironment.  As such, there are no comments on the work. However, all experiments are done on mice. Therefore, I would like to see the authors' reasoning regarding how the established patterns will correspond to a person.

I recommend publishing your work.

Response:

We appreciate the reviewer's suggestion. In this study, the influence of supplementation of SFBS on gut microenvironment was validated in the mice model and clinical trial. Throughout the results, the upregulated activity toward the production of 4-aminobutanoate, 1-amino-2-propanol O-2-phosphate, and catechol, was identified in the gut microenvironment of recruited participants and mice model with SFBS supplementation (Table 6 and 4). These results first suggest the similar influence of SFBS supplementation on manipulating gut microenvironment. Presently, the biochemical variables, including blood sugar, total cholesterol (TC), and triglycerides (TG), are continuously monitored in distinct populations of enrolled participants with or without SFBS supplementation. These results may be practicable for establishing specific patterns that correspond to humans.

Reviewer 2 Report

With the knowledge that diets high in fats or sugar are demonstrated to be risk factors for numerous chronic diseases, this study “Influence of symbiotic fermentation broth on regulating metabolism with gut microbiota-metabolite profiles is estimated using third generation sequencing platformrevealed that the diet addition of soybean fermentation broth containing saponin reduced hyperlipidemia and hyperglycemia, metabolic disorders associated with high-fat-fructose diets, through alterations in gut microbiota, their metabolites, and downstream pathways.

The topic is important and the manuscript provides a comprehensive analysis of the subject. In summary, the title stresses the value of the study; the abstract includes sufficient information to stand alone; the introduction summarizes the topic current state and knowledge in the field and explains why the experiment was needed; the methods are distinctly described with adequate details; the results are accurately presented, with relevant data given in tables and figures; in the discussion chapter the findings of the study are logically explained, followed by limitations of the study; the conclusions support and clearly summarize the experiment.

I would recommend this manuscript after the following suggestions have been attended to:

- a few typos and minor details should be attended to (e.g.,Table 4, line 240)

- Discussion: in this section authors should correlate and compare the results of the study with recent findings and previously published studies in the field, such as doi: 10.1016/j.foodres.2022.112284; doi: 10.1016/j.clnu.2020.04.005; doi: 10.3390/nu12010222

Minor editing

Author Response

Response to Reviewer 2's comments

With the knowledge that diets high in fats or sugar are demonstrated to be risk factors for numerous chronic diseases, this study “Influence of symbiotic fermentation broth on regulating metabolism with gut microbiota-metabolite profiles is estimated using third generation sequencing platform” revealed that the diet addition of soybean fermentation broth containing saponin reduced hyperlipidemia and hyperglycemia, metabolic disorders associated with high-fat-fructose diets, through alterations in gut microbiota, their metabolites, and downstream pathways.

The topic is important and the manuscript provides a comprehensive analysis of the subject. In summary, the title stresses the value of the study; the abstract includes sufficient information to stand alone; the introduction summarizes the topic current state and knowledge in the field and explains why the experiment was needed; the methods are distinctly described with adequate details; the results are accurately presented, with relevant data given in tables and figures; in the discussion chapter the findings of the study are logically explained, followed by limitations of the study; the conclusions support and clearly summarize the experiment.

I would recommend this manuscript after the following suggestions have been attended to:

- a few typos and minor details should be attended to (e.g.,Table 4, line 240)

Response:

In addition to the indicated typos, the spelling of the revised manuscript is carefully checked prior to submission.

- Discussion: in this section authors should correlate and compare the results of the study with recent findings and previously published studies in the field, such as doi: 10.1016/j.foodres.2022.112284; doi: 10.1016/j.clnu.2020.04.005; doi: 10.3390/nu12010222

Response:

The relevance between our work and related studies was further compared and discussed in the revised manuscript according to the reviewer's comment (please see lines 285-298; Ref. 21 and 22).

Reviewer 3 Report

The article is original and very relevant for the field. The authors supplemented the soybean fermentation broth containing saponin (SFBS) in male C57BL/6 mice fed high-fat-fructose. In addition to the diminishment of weight gain, the SFBS reduced hyperlipidemia and hyperglycemia. Also the supplementation of SFBS reprogrammed the high-fat-fructose diet-related microbial community with a relatively high abundance of potential probiotics.

Finally the study suggest that SFBS may function as a nutritional supplement for people with diverse metabolic disorders associated with high-fat-fructose diet.

Next studies in duplicate would be very usefull.

The conclusions are consistent with the evidence and arguments presented. The results of the study offer further evidence that continued research in the area of nutritional supplements for people with diverse metabolic disorders.

The references are appropriate, including some relevant authors experience in the field.

I recommend some minor corrections.

1.     Line 121 OTUs- correct is OUTs; line 215 OTUS correct as OTUs

2.     All the text should be aligned JUSTIFY

3.     References should be written according to Instructions for authors. Addition of doi code of the articles would be usefull.

Author Response

Response to Reviewer 3's comments

The article is original and very relevant for the field. The authors supplemented the soybean fermentation broth containing saponin (SFBS) in male C57BL/6 mice fed high-fat-fructose. In addition to the diminishment of weight gain, the SFBS reduced hyperlipidemia and hyperglycemia. Also the supplementation of SFBS reprogrammed the high-fat-fructose diet-related microbial community with a relatively high abundance of potential probiotics.

Finally the study suggest that SFBS may function as a nutritional supplement for people with diverse metabolic disorders associated with high-fat-fructose diet.

Next studies in duplicate would be very usefull.

The conclusions are consistent with the evidence and arguments presented. The results of the study offer further evidence that continued research in the area of nutritional supplements for people with diverse metabolic disorders.

The references are appropriate, including some relevant authors experience in the field.

I recommend some minor corrections.

  1. Line 121 OTUs- correct is OUTs; line 215 OTUS correct as OTUs

Response:

In addition to the indicated typos, the spelling of the revised manuscript is carefully checked prior to submission.

  1. All the text should be aligned JUSTIFY

Response:

The format of the text is revised according to the reviewer's comment prior to submission.

  1. References should be written according to Instructions for authors. Addition of doi code of the articles would be usefull.

Response:

The doi code of the cited reference is provided in the revised manuscript.
